# Gender differences in perceptions of "joint" decision-making about spending money among couples in rural Tanzania

Ibukun Owoputi[1]*, Rosemary Kayanda[2], Rachel Bezner Kerr[3], Juster Dismas[2], Prosper Ganyara[2], John Hoddinott[1,3,4], Katherine Dickin[5]

1 Division of Nutritional Sciences, Cornell University, Ithaca, NY, United States of America, 2 IMA World Health, Mwanza, Tanzania, 3 Cornell University, Department of Global Development, Ithaca, NY, United States of America, 4 Charles H. Dyson School of Applied Economics and Management. Cornell University, Ithaca, NY, United States of America, 5 Department of Public and Ecosystem Health, Cornell University, Ithaca, NY, United States of America

* io57@cornell.edu

**Data Availability Statement:** The Tanzania Food and Nutrition Centre (TFNC) hold the ownership rights to these data. Researchers wishing to access

## Abstract

Family and cultural contexts can constrain the effectiveness of evidence-based interventions designed to improve the health and wellbeing of women and their children. Unequal power relationships within the household may underlie the failure of many programs targeting women to achieve their intended impact. To reduce these unequal power dynamics within the households, many programs or interventions aim to both assess and improve the gender dynamics between husbands and wives within the household. Decision-making is one important facet of these dynamics and has been linked to health outcomes for women and children. However, household decision-making is rarely observed and often difficult to capture. This study aimed to use qualitative research to further understand one aspect of decision-making, namely on how to spend money. In two regions of Tanzania, we used surveys and interviews to explore different perspectives on spending and allocation of resources among 58 couples in rural farming households. While many men and women initially reported that they made decisions jointly, most women stated they would often concede if there was a disagreement or argument around spending. These results highlight the different perceptions of joint decision-making between men and women. We compared these results to survey responses on decision-making and found differences within and between couples across interview and survey responses. Based on the differences in qualitative and survey responses within couples and how they reported dealing with disagreement, our study found households were on a spectrum from no cooperation in decision-making to full cooperation. Our results highlight challenges for assessing decision-making on spending and ultimately improving these decision-making dynamics within the household. These challenges are especially important for maternal and child behavioral change and provide insights on why many interventions aimed at improving women's decision- making power on money may not reach their full potential.

the data must contact TFNC (info@tfnc.go.tz or https://www.tfnc.go.tz/contactus).

**Funding:** KD served as the PI for the award from United Kingdom Department for International Development (DFID) https://www.gov.uk/government/organisations/department-for-international-development. This award was subcontracted to IMA World Health. The funders had no role in study design, data collection and analysis, decision to publish, or preparation of the manuscript.

**Competing interests:** The authors have declared that no competing interests exist.

## Introduction

Households and family systems are situated within greater social, economic, and political environments, and they play an important role in behavioral change interventions [1–4]. Understanding the complex relationships among people within the household is important for understanding the context of interventions [5]. Households have often been considered a "black box" for power and gender dynamics within the household, presenting challenges for interventions operating at the household level.

Addressing these household processes and dynamics by exploring how household decisions are made can enhance the effectiveness of programs and policies. It is widely understood in the global health and development community that ignoring gender dynamics within the household can hinder development objectives, as these dynamics can affect the ability of women to practice recommended health and nutrition behaviors [6–12]. The Sustainable Development Goals include a goal explicitly focused on improving gender equality [13]. Despite the acknowledgement that it is important to improve gender equality, defining and conceptualizing it remains a challenge.

Gender equality has been operationalized in many ways. Gender equality is difficult to measure, and many different concepts had been developed to help capture different aspects of gender equality. Examples of these concepts include **women's empowerment**, or "the processes by which those who have been denied the ability to make choices acquire such an ability" [14] **bargaining power**, the "threat point", or the "fall-back position" which is how well off a person would be without cooperating within a household [15] and **autonomy or agency**, or "people's ability to use those capabilities and opportunities to expand the choices they have and to control their own destiny" [16]. These concepts all include **decision-making**, which is the ability to negotiate allocation of resources and income [8, 17–19]. Decision-making is often at the core of understanding household dynamics and is important for improving gender equality. While improving a woman's ability to make decisions on her own is important in its own right, a review by Carlson et al. across many different countries in the Global South found that it is also important for improving childhood nutritional status [8]. Women's decision-making power has generally been linked to better long- and short-term health outcomes for both women and children across many different countries in Sub-Saharan Africa [8–12].

Although many different types of household decision-making are important for health outcomes, resources controlled by women are usually associated with more expenditures on children [7]. For example, while mothers may know what foods to feed their children, their practices may be restricted due to lack of decision-making authority or access to budget [20]. The effectiveness of Social and Behavioral Change (SBC) interventions to improve women's and child's dietary diversity may be constrained if they are aimed at mothers and overlook the role men play in making decisions on spending money to improve maternal, infant, and young child health behaviors [21, 22].

Multiple indicators, methods, and proxy measurements have been developed to capture decision-making within the household. For example, a commonly-used index for measuring women's empowerment, Women's Empowerment in Agriculture Index (WEAI), includes interviews with both men and women to assess inequality around several concepts within the household, including decision-making around spending money, control over income, and resource allocation [6]. Several survey measures assess whether decisions are made alone or jointly and are widely used in surveys such as the Demographic and Health Surveys (DHS). Women's decision-making is often captured in national level surveys, such as the DHS, as a simplified measure of empowerment that can be linked to health behaviors and outcomes.

These questions are usually only asked of women respondents in a household, and include decision-making on a range of issues including household purchases [23].

Despite progress, decision-making continues to be both difficult to measure and contextually-variable. For example, although a woman may make the decision in a certain area, she may still be operating within the confines of her husband's preferences [9]. In addition, an increase in the amount of sole decision-making by the woman or joint decision-making by the couple is often seen as an indicator of improvement in household-level gender equality, but women's sole and joint decision-making are not synonymous [24]. Even in surveys such as the DHS, women are considered to have control over their income, assets, or participation in decision-making if they make the decisions jointly with their husbands or alone [25]. Interpretation of joint decision-making versus sole decision-making, however, can differ across contexts, type of decision, and even the nature of the couple dynamics.

Decision-making is difficult to observe and commonly assessed through women's self-reporting, focusing on one spouse's perceptions of decision-making and assuming that this represents what is happening within a household [26]. Some researchers who have studied both spouses have found that spouses may disagree over who makes the decisions or owns assets within a household, and men may be more likely to claim sole ownership [26, 27]. The association between women's decision-making and health outcomes may be underestimated when only the woman's account is considered, as agreement between couples on the wife's decision-making may have a stronger association with health care decisions or health outcomes [28–30]. The few studies focused on differences in decision-making perceptions between couples in the global south, mostly in south Asia (with one in Guatemala), found that couples often disagree on who makes decisions, has the autonomy, or control over the household resources [28, 29, 31, 32]. In addition, most studies in south Asia have focused on decision-making related to family planning and health care. In a study of farming households in Tanzania, researchers found discrepancies between husbands' and wives' perspectives of who makes different household decisions [26]. Previous studies found wide discrepancies within couples on accounts of decision-making [29, 33]. In addition, couples may have a different understanding of who has the final say, which is often assessed in decision-making surveys [29, 33]. Despite growing acknowledgement that it is important to include differing viewpoints from men and women, little research explores how these differing viewpoints affect health and nutrition outcomes, particularly in different contexts [24, 26, 30, 34].

Although capturing decision-making using qualitative methods can contextualize findings and capture nuance, there are few such studies [19]. Qualitative research can give further insights into how decision-making is perceived, across different cultural settings and between men and women. Decision-making on spending money in the household is important for a wide range of development outcomes that depend on the generation and use of household income.

This paper aims to deepen understanding on one aspect of decision-making, specifically household by comparing the perspectives between men and women within the same household on decision-making around spending money and allocating resources. We discuss qualitative interviews exploring differences in perceptions of decision-making around spending money in couples in rural Tanzania and married. We also compare interview responses to commonly used DHS survey questions assessing decision-making on spending money and resource allocation. This study highlights implications for measurement of decision-making in interventions designed to improve maternal and child health and wellbeing.

## Methods

This research was part of a larger project aimed at reducing childhood stunting in 5 regions of Tanzania. We conducted this research in 2 of 5 regions included in the larger project: Kagera and Shinyanga, predominantly rural regions with rates of maternal and child mortality among the highest in Tanzania [25]. The study described in this paper was designed to further understanding of the social and gender influences that affect decision-making on spending money in rural, farming households. In rural Tanzania, low education rates often restrict economic opportunities, contributing to concentrations of poverty. Education is lower for women, with only 4% of women in rural areas completing secondary school [25]. Four-fifths of the poor in Tanzania (those living on less than US $1.9 per day) live in the rural areas [35] and most earn their income and livelihoods from agricultural work [25].

Data collection included couples' interviews and surveys on decision-making on money. Survey data also included demographics. During the analysis, we saw differences between the survey responses and interview responses within the same household for decision-making on spending money. This difference emerged as a major inductive theme and is the focus of this paper.

### Sampling

Prior to beginning this study, survey data on household characteristics and demographics were collected for every household containing a pregnant woman or a mother of a child under the age of 2 in each of our 35 study villages across the two study regions (Kagera and Shinyanga). For this study, we sampled 14 villages (7 per study region) from the 35 villages in the previous study. We excluded villages that were more than a 2.5-hour drive from the research team's lodging site, and purposefully sampled villages to ensure diversity in religion and population size.

From the 14 villages, purposeful sampling of women was done to ensure maximum variation in types of women and households (e.g., women's age, polygamy, and household size) [36]. We excluded households that did not speak Swahili because staff members were unable to conduct the research in the local languages, and we did not have access to reliable translation of those languages. In each village we randomly chose a total of 8 potential women for interviewing and collaborated with village and community leaders by giving them the ranked list of women. These leaders contacted the women and men in these households until we secured 4–5 households for that village, as some households were unavailable or had moved. The local leaders then prepared those households for our visit to conduct the interviews, using the wording in the consent form. We continued until we reached theoretical saturation [37] which occurred after interviewing 58 households (58 men + 58 women).

### Data collection

We completed the research study from March 4 to June 3, 2019. Four research assistants (RAs), two men and two women, conducted the data collection. All RAs were college educated and had been trained in qualitative methods. RAs were observed during practice interviews with community members prior to data collection to ensure reliability.

Female RAs interviewed female participants, and male RAs interviewed male participants. Our previous research demonstrated that participants were more comfortable completing interviews with same-gender research assistants [38]. Husbands and wives were interviewed at the same time, in private locations in different parts of the compound.

There were 2 days of data collection per household. On day 1, pairs of RAs visited households to introduce themselves, explain the research, and ask if the couple wished to participate. After couples agreed to participate, they were separated to complete the detailed consent

process. RAs then conducted 30–45-minute quantitative surveys using tablets to collect information on demographics (e.g., age, literacy) and decision-making. The subset of the survey results analyzed for this paper included DHS decision-making questions on spending money, specifically "Who decides how the money you earn will be used?", "Who usually decides how your partner's earnings will be used?" and "Who usually makes decisions about making major household purchases?" The responses for these questions included "Respondent", "Partner", "Jointly", or "Other (Specify)". This 1st day of data collection was used to establish rapport with the interviewees.

On the second day of data collection, the same RA who surveyed each participant conducted the qualitative interview. Participants were asked how they made decisions on spending money in their household (whether by themselves, with their partner, or their partner decides alone by him/herself). Participants were also asked how spending decisions were normally made in the household, intimate partner violence or conflict about spending money, situations where disagreement around spending arises, who has the final say, and if other household members are normally involved in these types of spending decisions. The interview took 1–1.5 hours per participant. Interviewers took detailed notes after each interview, which were discussed in detail with the research team. Each interview was recorded and transcribed by the respective interviewers. The transcribed Swahili interviews were then reviewed by our research coordinator and sent to an independent party for translation in English. After translation, each English transcript was reviewed twice: once against the audio recording by the original interviewer and a second time by a different team member.

## Ethical approval and consent

This study was approved by National Medical Research Coordinating Committee of the Ministry of Health and Social Welfare (PROTOCOL NIMR/HQ/R.8a/Vol. IX/2905) in October 2018 and the Cornell University Ethics Review Board (Protocol #1611006801) in July 2018. Informed written consent was obtained from all participants prior to beginning any data collection.

## Data analysis

We used Stata/SE v15.1 (StataCorp, College Station, TX) to summarize survey data (demographics, household, and DHS decision-making questions on spending money). For the decision-making questions, we followed prior research in assessing concordance and discordance [30]. Couples were categorized as "concordant" if they provided the same response, and "discordant" if they provided different responses. For example, a couple where the wife stated that she decides how the money that she earns is used, yet the husband states that he decides how the money his wife earns is used were considered "discordant". If both the husband and wife agreed that the husband decides the money the wife earns will be used, that was categorized as "concordant".

Transcripts were coded and analyzed using qualitative software NVivo 12 by 7 research team members, including 3 members of the data collection field team. The analysis used a mixed deductive and inductive approach [39]. For the deductive analysis approach, we used the interview questions to guide the development of some of the coding. We also used open coding, which allowed for codes, concepts, and themes to emerge inductively from the data [40]. The codebook was initially created through open coding by the 3 team members who had been in the field and completed the first round of coding for all interviews. After the first round of coding was completed, 4 additional team members coded each transcript a second time. After the second round of coding, all transcripts were finalized and reviewed by a 3rd

person. Throughout the entire coding process, we continued to discuss and refine the code-book as an iterative process. After coding each interview, we completed summaries for each couple to compare responses between husbands and wives within the same household, highlighting differences in perceptions of decision-making and how couples respond to disagreements.

We used the qualitative data to understand these gender differences in perceptions of decision-making on spending, and to understand what is meant by both "joint" and "sole" decision-making. The interviews also explored what happens in conflict, and who has the final say. These interviews helped us understand and give context to the DHS decision-making questions. This was completed by comparing the survey DHS results to the interview responses from the same couples on decision-making on spending money. We categorized households based on these differences and found that they fell on a spectrum of decision-making.

## Results

### Descriptive characteristics

Demographic characteristics of the households are found in Table 1. Across the 58 couples, husbands were on average 8 years older than their wives. The highest completed education level for participants was primary school, and one-third of men and women were not fully literate. As is common in the study site, many couples lived with the husband's mother or parents. On average, couples had been living together 8 years, and in just under 20% of couples, the man had multiple wives. We interviewed different wives to get a range of polygamous couples; about half the couples included the husband and his first wife, about a third of couples

**Table 1. Study demographics for male and female participants.**

| Variable | Men [mean ± SD or n (%)] | Women [mean ± SD or n (%)] |
|---|---|---|
| | N = 58 | |
| Age | 34.9 ± 11.5 | 27.2 ± 7.3 |
| Age Difference (Men-Women) | 7. 7 ± 8.7 | |
| Household Size | 5.8 ± 2.7 | 6.4 ± 3.0 |
| Education | | |
| Pre-Primary | 7 (14.3%) | 9 (17.3%) |
| Primary | 37 (75.5%) | 38 (73.1%) |
| Secondary 'O' Level | 3 (6.1%) | 3 (5.8%) |
| Post-Secondary 'O' Level | 1 (2.0%) | 2 (3.9%) |
| Post-Secondary 'A' Level | 1 (2.0%) | 0 (0.0%) |
| Literacy | | |
| Cannot read at all | 9 (15.5%) | 14 (24.1%) |
| Able to read partially | 7 (12.1%) | 5 (8.6%) |
| Able to read fully | 42 (72.4%) | 39 (67.2%) |
| Years Living Together | 8.33 ± 6.6 | 8.26 ± 6.5 |
| Years Married | 8.12 ± 6.2 | 7.91 ± 6.6 |
| Polygamous | 10 (17.2%) | 11 (19.0%) |
| Rank | | |
| 1st Wife | | 6 (54.6%) |
| 2nd Wife | | 4 (36.4%) |
| 3rd Wife | | 1 (9.1%) |
| Lives with Mother-in-Law | 11 (19.0%) | 35 (60.3%) |
| Lives with Mother | 33 (56.9%) | 9 (15.5%) |

included the husband and his second wife, and one was the husband and his third wife (each wife in a polygamous couple considered herself and her husband a separate household). When interviewing a polygamous husband, we specified which wife we wanted him to refer to when describing decision-making.

It was notable that within couples, women and men often differed in reporting on factors such as household size, polygamy, and living arrangements. Within the same couples when asked about polygamy, 10 men said they had multiple wives while 11 women said that their husband had multiple wives. Some differences may be due to social desirability bias, but in other cases, couples had different perspectives. For example, the husband's mother might live next door, and while the husband considers them all to be living together, the wife might consider them to be separate households.

## Quantitative decision-making results

In the quantitative data on decision-making around spending using the standard DHS questions, the most common response for men was "joint" (43.1%) while half of women responded that the "husband decides" (Table 2). None of the men or women in our sample reported that women were the primary decision-makers for major household purchases. In addition, most

**Table 2. Survey results from DHS questions on decision-making around spending money.**

| Survey Question | Female Response | Male Response |
|---|---|---|
| | N (%) | N (%) |
| Who decides how the money you earn will be used? | | |
| Wife | 10 (17.2%) | 0 (0%) |
| Husband | 29 (50.0%) | 25 (43.1%) |
| Jointly | 18 (31.0%) | 33 (56.9%) |
| Doesn't Know | 1 (1.7%) | 0 (0.0%) |
| Who usually decides how your partner's earnings will be used? | | |
| Wife | 2 (3.5%) | 19 (32.8%) |
| Husband | 37 (63.8%) | 9 (15.5%) |
| Jointly | 19 (32.8%) | 30 (51.7%) |
| Who usually makes decisions about making major household purchases? | | |
| Wife | 0 (0.0%) | 0 (0.0%) |
| Husband | 46 (79.3%) | 17 (29.3%) |
| Jointly | 8 (13.8%) | 40 (69.0%) |
| Other (Parents or in-laws) | 4 (6.9%) | 1 (1.7%) |
| Couples Agreement (comparing answers to questions above) | | |
| Who decides how the money you earn will be used? (Women's Income) | | |
| Concordant[1] | 17 (29.8%) | |
| Discordant[2] | 40 (70.2%) | |
| Who decides how the money you earn will be used? (Men's Income) | | |
| Concordant[1] | 28 (48.3%) | |
| Discordant[2] | 30 (51.7%) | |
| Who usually makes decisions about making major household purchases? | | |
| Concordant[1] | 19 (32.8%) | |
| Discordant[2] | 39 (67.2%) | |

[1]Concordant: Within a couple, the woman and man gave the same response
[2]Discordant: Within a couple, the woman and the man gave different responses

couples disagreed on who made the decisions around income and major household purchases in the quantitative survey responses.

## Qualitative decision-making results

In the beginning of the interviews, participants were asked how they normally made decisions around spending money: Together as a couple, by themselves, or their partner by him/herself. (Table 3). There was more concordance in these responses, as the most common initial response for male and female respondents was that they made decisions together/jointly.

We found, however, that later in the interview when asked about what happens during arguments or conflict, what was referred to as "joint decision-making" was defined differently within and between couples. Men were more likely to talk about hiding money or spending without involving their wives even though they reported making joint decisions, and it was acknowledged by both men and women that men usually had the "final say".

Analyzing the differences in qualitative responses within couples and how they reported dealing with disagreement, we found that households were on a spectrum from no cooperation in decision-making to full cooperation (Fig 1). On the right end of the spectrum, couples reported discussing all spending and did not move forward on any decision unless both agreed ("true cooperators"). On the left end of the spectrum, couples usually both agree that the husband was the one who made all the decisions, reported in household decision-making around spending ("absent communication or coordination"). While almost all female and male participants agreed that they made household spending decisions jointly, female participants often conceded if there was a disagreement or argument around spending. Most households in our sample fell in that central area of the spectrum ("Conceders & Dominators"). This decision-making style was common in polygamous couples as well, as these types of couples often reported they made decision together jointly, but the wife later stated that she would concede in disagreements.

In the following section and Tables 4–6, we present quotes from the 5 example households shown along the spectrum in Fig 1, illustrating similarities and differences with specific examples of how couples described decision-making processes. Although we originally aimed to create a typology of different types of decision-making households, we found that households did not fit neatly into discrete categories. Couples differ in how they make decisions depending on the timing of the decision, and the type or magnitude of the decision, and they do not fall into the same typology consistently. Therefore, we use these example couples to highlight important themes across the spectrum of how joint decision-making is perceived. These examples represent the most common situations, but some households also included in-laws or other family members in decision-making.

**Table 3. Couple's decision-making interview results.**

| Response to decision-making ("When you make decisions on spending money in your household, how do your normally decide: 1) together when you are with your husband/wife, 2) by yourself or 3) your husband/wife by himself/herself?") | Female Response N (%) | Male Response N (%) |
|---|---|---|
| Husband only | 6 (10.3%) | 6 (10.3%) |
| Joint or together | 49 (84.5%) | 52 (89.7%) |
| Other (Separate finances or parents/in-laws) | 3 (5.2%) | 0 (0.0%) |
| Couples Agreement | | |
| Concordance | 44 (75.9%) | |
| Discordance | 14 (24.1%) | |

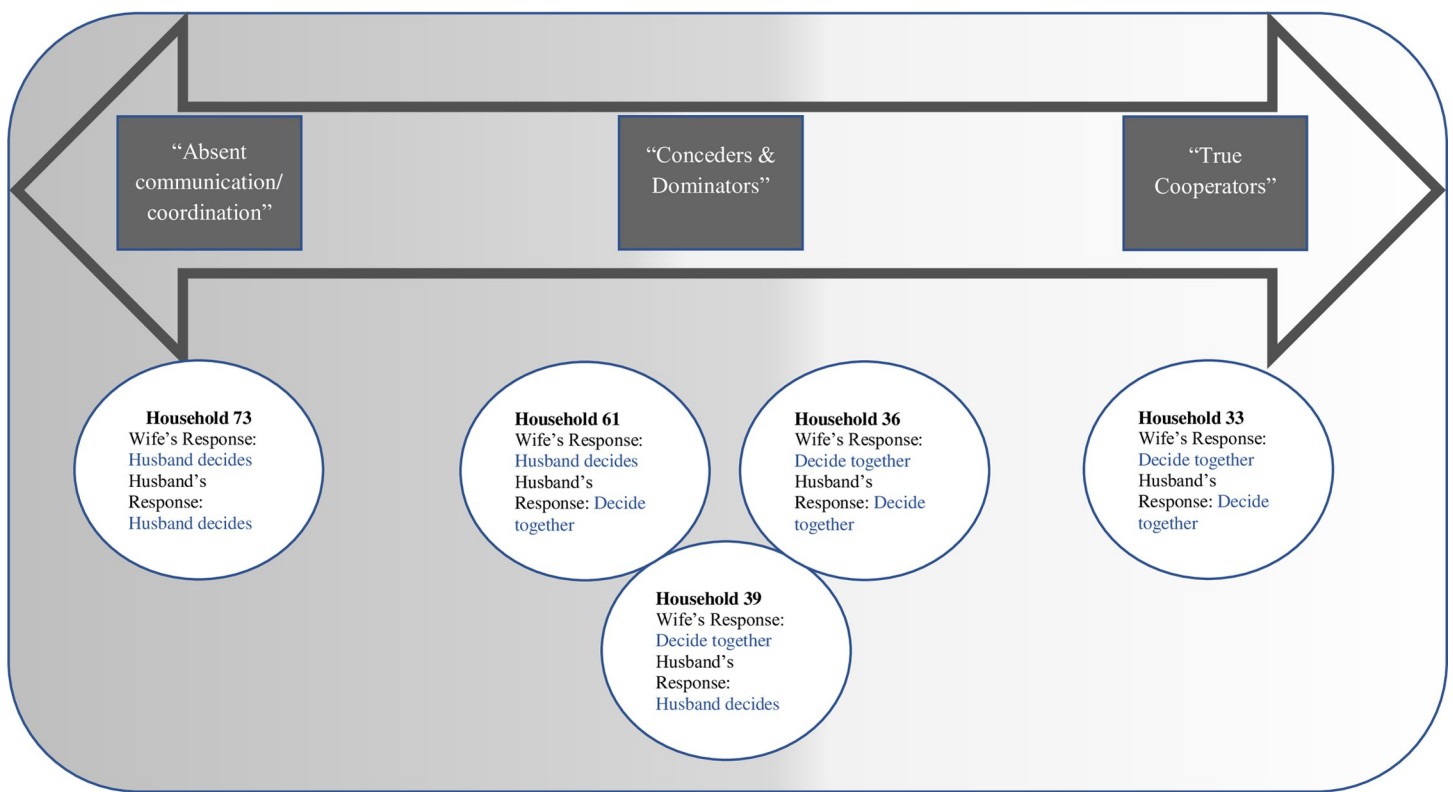

**Fig 1. Spectrum of decision-making: Example couples whose responses illustrate the variation in perspectives on who makes decisions on spending money.** White circles represent where example households fall across the spectrum, and include their response to 1ˢᵗ interview question asking if decisions are made together, if the husband decides alone, or if the wife decides alone. Spectrum arrow illustrates types of responses to interview questions on how they make decisions during disagreements, ranging from "Absent Communication/coordination to "True Cooperators" (Example quotes from these couples are detailed in Tables 4–6).

**Household A: Both partners say the husband is the sole decision-maker.**   In household A neither partner reported joint decision-making. We classified these types of households as "Absent communication/coordination" households. Few households fell on this end of the

**Table 4.  Decision-making for household A: Response to decision-making ("When you make decisions on spending money in your household, how do your normally decide: 1) together when you are with your husband/wife, 2) by yourself or 3) your husband/wife by himself/herself?") and response to disagreements.**

| | "Absent communication/coordination" | |
|---|---|---|
| | **Wife- 32 years old** | **Husband- 50 years old** |
| **Example responses to how decisions are made** | "When my husband gets money, he usually decides alone what to buy." | "There are very few things that we cooperate on, but with serious issues I make the decisions alone because she said she is not capable." |
| **Example responses to how disagreements are handled** | "He doesn't tell us what he wants to buy, he is the one who has the decisions, if it is about the cows, he would just go to buy them then we would only see the cow has been bought." | "You know in our village, our women lack education that is a big problem. The person who makes the decision on what to buy is the father of the family, in case the father disappears also know that the family is lost, you cannot depend on the mother to help when the father has no money. Such a thing cannot happen, the person with more power is the father, he is the light to move forward or backwards, but the mother has nothing to offer because she lacks education." |

**Table 5. Decision-making in household B and household C: Response to decision-making ("When you make decisions on spending money in your household, how do your normally decide: 1) together when you are with your husband/wife, 2) by yourself or 3) your husband/wife by himself/herself?") and response to disagreements.**

| | "Conceders and Dominators" | | | |
|---|---|---|---|---|
| | Household B | | Household C | |
| | Wife- 33 years old | Husband- 38 years old | Wife- 33 years old | Husband- 38 years old |
| Example responses to how decisions are made | "When he gets money, he is the one who decides." | "The important one that we use is the one of working together." | "We usually decide together." | "We usually use all of them but mainly it's a man decision it's very rare for other categories to be used." |
| Example responses to how disagreements are handled | "If he sells rice and decides alone, the woman has no say even if you disagree, if he wants to sell then he will sell... the woman has no say at all, the man would say 'can the woman really tell me what to do with the money?'" | "It happened in 2006 I decided alone, we farmed cotton but later I thought that I should decide otherwise and do business without involving my wife, without knowing I did that business without thinking and it wasn't within my plans and my capital was small, I spent the money irresponsibly and I didn't achieve anything. After I ran out of money my wife told me that 'do you see the impact of doing business without telling me?'... I felt bad and since then I learned that everything I want to do I must involve her" | "You know when a man has a money there are a lot of people who depend on him... sometimes we have to let the man make final decision whether you like it or not" | "I will just inform her that I have a plan to do this, what do you think, then she might say do it but even if she would say no its very rare for me to take her opinions because she has no authority to refuse what has been planned" |

spectrum. This is illustrated in the quotes below from a couple exemplifying this situation (Table 4). When asked "When you make decisions on spending money in your household, how do your normally decide: 1) together when you are with your husband/wife, 2) by yourself or 3) your husband/wife by himself/herself?", couple A gave the responses below in Table 4, indicating that the husband makes most of the decisions on his own. When probed about what happens during a disagreement or argument, the wife talked about how her husband does not involve her in any household spending at all, she only sees the household purchases once her husband brings them home. The husband also stated that because women have a "low level of education" they are not capable of contributing to any household decision-making. In this

**Table 6. Decision-making in household D and household E: Response to decision-making ("When you make decisions on spending money in your household, how do your normally decide: 1) together when you are with your husband/wife, 2) by yourself or 3) your husband/wife by himself/herself?") and response to disagreements.**

| | "Conceders and dominators" | | "True Cooperators" | |
|---|---|---|---|---|
| | Household D | | Household E | |
| | Wife- 37 years old | Husband- 60 years old | Wife- 25 years old | Husband- 38 years old |
| Example responses to how decisions are made | "We discuss with my husband, you know for us here everyone has savings so we sit down and agree on what to do." | "We have to decide together." | "We sit down and discuss what we should buy, what we should do. We cooperate with each other on what we should do to advance." | "We both plan with each other." |
| Example responses to how disagreements are handled | "I was asking about why he was spending money recklessly because we had sold some of our harvest and he was spending the money recklessly. When I asked him, he became angry and he beat me up." | "She trusts me because all the spending I do she doesn't see, she hears I took 3 kg of meat yet she doesn't see it at home, or she may hear I had some money, let's say one hundred thousand shillings, yet she doesn't know how it was used." | "It is really great when you are making decisions together at home, even problems cannot arise in the family thereby you trust each other, you do things together so when he gets a problem you help him and he helps you, not family exclusion, we don't have that." | "She has the right [to buy things without telling me] but it's the closeness that she and I have or cooperation, that is why she involves me." |

household, the husband and wife have the same level of schooling (both completed primary school).

**Household B & Household C: Only the husband OR only the wife describe decision-making as joint.**   Some couples fell more towards the middle of the spectrum, and we classified them as "Conceders and dominators." In some couples, only one person reported that they made joint decisions on spending and the women revealed later in the interview that she would concede when disagreements/arguments arose (Table 5). In household B, only the husband stated joint decision-making when asked initially in the interview. When probed about what happens during a disagreement or argument, the same participants in household B gave the responses below (Table 5). The husband in household B discusses how they used to not communicate or coordinate, but he later realized the importance of involving his wife in household decisions. He describes this change as an improvement, as he considers telling his wife what he is going to do as "involvement", or "joint" decision-making. In contrast, the wife in household B says that as a woman she cannot challenge her husband's spending decision. The wife in this case does not consider their situation as joint decision-making because she feels she cannot go against her husband for any decision.

In household C, only the wife stated joint decision-making when asked in the interview (Table 5). The husband in household C makes the decisions on household spending on his own, similar to the previous household. While both households seem to make spending decisions in a similar way (with the wife having no authority to contest the man's decision), the wife in household C reports that decision-making is joint because her husband at least informs her of his decisions while the wife in household B did not consider her husband informing her to be joint decision-making. While the wife in both households would concede to the husband, the husband in household B and the wife in household C considered that joint decision-making.

**Household D & household E: Both the husband and wife describe decision-making as joint.**   Table 6 provides two examples of couples in which both partners initially reported making joint decisions on spending. In household D both the male and female participant initially said they made decisions together jointly, but if there was an argument or disagreement the woman would concede ("Conceders and dominators"). In this part of the spectrum, similar households also varied, in which some women would speak up if they disagreed but then concede to any pushback and other women would refuse to speak up at all.

By contrast, the couple in household E gave similar initial responses to household D, but when probed about what happens during a disagreement or argument, they appear to cooperate in decision-making around spending ("True Cooperators") (Table 6). While both the husband and wife in household E talk about the importance of togetherness and cooperation ("True Cooperators"), the husband in household D discusses hiding money from his wife because he believes she "trusts" how he spends money ("concedes and dominators") while his wife reveals concerns of intimate partner violence if she challenges her husband. In other households, some women would report some aspect of being afraid or unable to challenge their husband, either from fear of retribution or as part of conceding to cultural norms even if they did not explicitly report intimate partner violence within the household.

## Discussion

Our qualitative study highlighted the gendered differences in perceptions of decision-making on spending between and within couples, contrasts in responses depending on how questions are asked and implications for measurement of decision-making. Programs aimed at improving women's empowerment or autonomy often seek to increase the prevalence of households

practicing joint decision-making. Our research suggests that simply asking whether there is joint household decision-making may not be a sufficient or accurate metric to assess women's empowerment or gender equity. Capturing more details on couple communication and response to disagreement is important when assessing decision-making and represents an area of exploration for programs trying to measure and improve women's autonomy through decision-making. Decision-making processes are important to consider when designing interventions to improve diet quality and spending on more costly foods or healthcare. In addition, understanding the power relationships within the household is important for designing interventions [41]. Having further insight into how intra-household decision-making affects the practice of recommended behaviors can aid in improving effectiveness of interventions aimed at improving health and wellbeing of members at the household level.

Increasing women's decision-making power as it is currently measured may not be enough to get at the core of household dynamics. Although decision-making questions are often used in surveys, it remains challenging to ensure that these decision-making questions capture actual household behavior [42]. There is considerable merit to complementing quantitative surveys with more in-depth qualitative work to contextualize and better understand responses to questions surrounding decision-making. Our interviews were useful for understanding the roles and processes involved in decision-making rather than the specific allocation of resources. The analysis of differences in responses between men and women helped to further understand decision-making and highlight nuances that are missed by a decision-making survey.

Our study had many strengths. We gathered data separately from both men and women, which allowed each participant to answer without worrying about what their spouse thought. Sometimes spouses offer different information individually than they would provide if they were together; they may not feel free to be as honest when they are interviewed together, or may have different perceptions of their shared lives [43, 44]. Couples interviewed separately are more likely to provide different accounts than they would if they were interviewed together [45, 46].

Collecting data from men also allowed for comparison. Research on household decision-making often look at the perspective of a single member of the household, however, couples often disagree about who makes the decisions in the household. While men in our study were more likely to report joint decision-making around spending than sole decision-making, other studies have found that men may be more likely to report sole decision-making than joint decision-making, not recognizing the role of their wife in the decision-making process [27, 32, 47–49]. These differences in perceptions can have implications for explaining household gender dynamics. A study by Annan et al. defined situations where the wife says she has more decision-making ability than what her husband reports as "taking power", where being "given power" is defined as situations where the husband reports that his wife has a higher decision-making ability than she reports for herself [50]. They found that when these types of discrepancies exist, women "taking power" was associated with greater health outcomes for women and children [50]. These detailed patterns are likely context specific, as decision-making may ultimately be affected by gender, cultural, and social norms. In the setting for our research, we found men may be more likely to report joint decision-making, unaware of or ignoring the power and control dynamics at play which may prevent a woman from actively participating in true "joint" decision-making.

We did not capture decision-making around all types of decisions, and the examples in this paper are just a few case studies to illustrate the variation. Decision-making processes can vary widely based on what the decision is about, or how important the decision is [51]. Decision-making around spending money is likely to vary based on the amount of money that is

involved and the magnitude of the decision. While we were able to interview a large number or men and women for a qualitative study, it is possible that we missed some different types of decision-making households. It is unlikely that every household makes decisions the same way each time, another area for further research. Some households differ in how they make decisions depending on the timing (for example, the husband may spend several months away for work), the type of decision (e.g., couples may decide on their own to buy various household items but may need to consult in-laws when deciding on spending for agriculture or livestock). We found that categorizing households, e.g., as "joint decision-makers", ignores the large amount of variation in decision-making that exists within couples and households.

Often, there is an interplay between conflict and cooperation that occurs in households, as different family members may have different preferences and expectations [7, 51]. In this study, we focused on decision-making between the husband and the wife. Although the majority of participants lived with parents or in-laws and the qualitative interviews did probe about the involvement of other family members, the majority of interviewees did not report this involvement or reported limited involvement. It is possible that other family members may be more involved in other types of decisions (other than decisions around spending money or allocating resources) or that this was not common in our particular context.

This research was conducted with households in rural villages in Tanzania, which may not be generalizable to other areas or countries in the Global South, as context is an important influence on decision-making. In addition, we focused specifically on decision-making about spending and resource allocation, which may not be generalizable to all types of decision-making. This research has, however, added to the literature by identifying nuances that may be missed in how decision-making is measured in quantitative surveys, and how perceptions on decision-making may differ between men and woman within the same household.

Further research is needed on whether it is important to ask about the final say in decision-making. Although almost all couples in our sample reported that the husband has the final say (suggesting this may be the dominant cultural norm), a few couples reported that neither the husband nor the wife had the final say. Research could explore what is meant by "final say" and whether that furthers understanding of the decision-making process within households. Many women in our sample considered the husband as the final say simply because he was the one who brought in all or most of the household income. More research is also needed on different types of decisions, how and if decision-making involves in-law, parents or other family members, and how decision-making on spending varies with different amounts of money.

Responses on how couples handle disagreement around spending, either before or after the decision was made, provided insight into how couples made decisions. It was rare for men and women to conceive of a joint decision-making scenario where each partner had equal say and valued communication and cooperation. Women may feel they need to concede in household decision-making to avoid retribution, such as intimate partner violence. If limited to a survey, incorporating questions about intimate partner violence in decision-making when there is argument or disagreement and who has the final say can help to better understand some of the hidden dynamics at play. These may be areas for further research on how to effectively capture this information within a decision-making survey.

Ultimately, the complexity and nuances highlighted in our findings have important implications for all research and programming that involves decision-making at the household level. Interventions focused on improving women's decision-making power or reallocation of household resources need to understand how decisions are made within the household.

## Conclusion

In this study in rural Tanzania, we found that couples' decision-making varied along a spectrum ranging from little or no discussion to full cooperation, even though most respondents initially reported making decisions jointly. Although women and men may report joint decision-making in a survey, actual practices differ due to household power structures and other cultural, social, and gender norms. Comparison of responses between men and women within the same household, in both survey and qualitative data, allowed deeper analysis of understanding decision-making and how perceptions of decision-making can differ between and within couples. To illustrate the variation and nuances in household decision-making patterns, we described a spectrum of these dynamics and used example households to illustrate differences within couples and between households. Along the spectrum, "joint" decision-making ranged from "forced" agreement to differing levels of accordance. While a few couples reported enjoying discussing and making decisions together and saw themselves as a team, most women reported that they would go along with their husbands' decisions to avoid arguing or being beaten, even if they disagreed. On the most collaborative end of the spectrum, some participants felt that they could not make the best decisions for their family unless they were able to come to an agreement with their spouse, while in many households, women reported that they either did not have a say in decision-making or they had little influence on decision-making if the man is adamant about his decision. Couples may consider a decision "joint" if the husband at least asks his wife for input, even if he does not consider her suggestions or even if he just informs his wife of his decision. Qualitative interview responses about how disagreements around spending were handled in the family provided further insight into the level of autonomy a woman had in the household. Ultimately, understanding these nuances can help uncover factors that influence behavioral change at the household level in community-level interventions.

## Author Contributions

**Conceptualization:** Ibukun Owoputi, Rachel Bezner Kerr, John Hoddinott, Katherine Dickin.

**Formal analysis:** Ibukun Owoputi, Rachel Bezner Kerr, Juster Dismas, Prosper Ganyara, John Hoddinott, Katherine Dickin.

**Funding acquisition:** Katherine Dickin.

**Investigation:** Ibukun Owoputi, Rosemary Kayanda, Juster Dismas, Prosper Ganyara.

**Methodology:** Ibukun Owoputi, Rosemary Kayanda, Rachel Bezner Kerr, Juster Dismas, Prosper Ganyara, Katherine Dickin.

**Project administration:** Ibukun Owoputi, Rosemary Kayanda.

**Supervision:** Ibukun Owoputi, Rosemary Kayanda, Katherine Dickin.

**Visualization:** Ibukun Owoputi.

**Writing – original draft:** Ibukun Owoputi, Rachel Bezner Kerr, Katherine Dickin.

**Writing – review & editing:** Ibukun Owoputi, Rosemary Kayanda, Rachel Bezner Kerr, Juster Dismas, Prosper Ganyara, John Hoddinott, Katherine Dickin.

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
