## [Decision Letter · Decision Letter 0]

18 Jul 2023

PONE-D-23-01170Gender differences in perceptions of “joint” decision-making about spending money among couples in rural TanzaniaPLOS ONE

Dear Dr. Owoputi,

Thank you for submitting your manuscript to PLOS ONE. After careful consideration, we feel that it has merit but does not fully meet PLOS ONE’s publication criteria as it currently stands. Therefore, we invite you to submit a revised version of the manuscript that addresses the points raised during the review process.

Two major concerns shown by the reviewers are: justification of small sample size and weak discussion. Authors must carefully attend them.

We look forward to receiving your revised manuscript.

Kind regards,

Muhammad Khalid Bashir, PhD

Academic Editor

PLOS ONE

Journal Requirements:

Reviewers' comments:

Reviewer's Responses to Questions

**Comments to the Author**

1. Is the manuscript technically sound, and do the data support the conclusions?

Reviewer #1: Yes

Reviewer #2: Yes

2. Has the statistical analysis been performed appropriately and rigorously? 

Reviewer #1: Yes

Reviewer #2: Yes

3. Have the authors made all data underlying the findings in their manuscript fully available?

Reviewer #1: No

Reviewer #2: No

4. Is the manuscript presented in an intelligible fashion and written in standard English?

Reviewer #1: Yes

Reviewer #2: Yes

5. Review Comments to the Author

Reviewer #1: Overall: this manuscript incorporates both a quantitative and qualitative exploration of household decision-making as part of a larger project in Tanzania. A comparison of spouses’ responses led to a robust discussion of the complexities of decision-making and the development of a spectrum that can be helpful in understanding household decision-making. Overall, this is an important topic and one that those working in gender are continuing to struggle with. At the same time, there are ways that the manuscript can be strengthened – from the framing in the introduction to specifics in the methods and results, to better contextualization of results and interpretation in the discussion – that would help to strengthen the manuscript overall. Please see more detailed comments below.

Abstract:

In the beginning part of the abstract as well as at the end, I wonder if it is more compelling to explain why, for health and wellbeing, it is important to explore dynamics around decision-making.

The second sentence of the abstract is run-on and it needs to be separated/revised.

Suggest removing “very” from sentences as it is a vague descriptor.

Based on what is written in the abstract, it is unclear how the complexities in decision-making led to the conclusion that couples/households are on a spectrum from no cooperation to full cooperation.

Introduction:

Paragraph 1: Can you expand/elaborate, give specifics on what conflict and cooperation about in households can look like?

Paragraph 1, line 4-5: Positive behaviors such as?

What development objectives does ignoring gender dynamics hinder?

Is the issue defining gender equality, or implementing and putting into practice and contextualizing what gender equality looks like?

Paragraph 2: women’s empowerment is related to gender equality, but distinct. I think the relationship between these concepts could be clarified. In addition, a lot of different concepts are covered in this paragraph, and how they relate, in particular how/why decision-making is important for improving gender equality, should be expanded.

I think the organization of the introduction could be improved to strengthen and improve clarity. I think it might be stronger to start the introduction making the important linkage between decision-making and health outcomes, which currently comes in paragraph 3, and why we should understand decision-making and then introduce key concepts. That might help the flow as the introduction currently goes back and forth between gender equality and decision-making. It might also be important to discussion the distinction between women’s decision-making and joint decision-making a bit earlier before page 4.

Page 5: “Research have found” – in what contexts? In addition, a lot of studies are summarized in this paragraph, and it might help to add a topic sentence or concluding sentence that summarizes/synthesizes the overarching key point in how these studies vary.

It might be helpful to, in the last paragraph of the introduction, contextualize the aims and clarify the location of the study.

Methods:

In the first paragraph, when contextualizing the larger project, it might be helpful to describe briefly the way that Phase A/B results were used as part of the project – were they formative, evaluative, etc.? Phase B focused on household-level attitudes and practices related to what exactly? What nutrition and health behaviors in particular?

It would be helpful to introduce the factors shown to be associated with decision-making in previous research earlier in the introduction if they were considered as part of sampling (e.g., age, education, etc.).

There is a lot of detail included about the overall study, but I wonder if detail about data collection that is not relevant to the current manuscript should be removed to focus specifically on the methods that led to the data analyzed here.

Data analysis approaches for quantitative data should be included in the data analysis section as well. How was concordance assessed?

In addition to the analysis of the qualitative transcripts, were particular approaches used to analyze the pile sort activity data? I know space considerations are important to consider, but the results seem to be important to include.

Results:

Suggest being specific in the presentation of results rater than saying “about.” For example, not “about 8 years” but presenting the specific number.

Any reflections on men’s reflections on DM with one wife vs. another?

p. 10, 2nd paragraph of results: this reads more like interpretation and should be better placed in the discussion.

Given the small sample size, I think presenting statistics with 1 decimal point seems more appropriate. This would make Table 1 consistent with Table 2.

The summary of concordance/discordance in Table 3 (and Table 4) could be elaborated in the narrative. Furthermore, it is important to look not only at whether they were concordant or not, but also what that concordant response was. Can that be broken out/disaggregated further? For example, were they concordance that men made the decision, that women mad ethe decision, etc.?

In describing the example households, it might be helpful to provide more background details about the household (perhaps from the demographic characteristics?). I would also perhaps develop a name for the households rather than using labels like “household 73.” In the tables with quotes from participants, the title of the tables seems to be the nickname/label that could be used. However, it’s important to make this consistent with the sub-heading title where you describe that/those households. Currently, they are slightly differently described.

While the tables with illustrative quotes are helpful, is there space to incorporate a few quotes into the main body of the results section?

Discussion:

One question that has come to mind in this analysis is that the framing is around household decision-making, but the analyses seem often focused on the husband and wife. Is there space to elaborate on the role of other family members in household decisions?

The first two paragraphs in the discussion could perhaps be synthesized with what is in the introduction so that the discussion dives right into the key findings and situating them within the context of existing literature and identifying programmatic implications.

Page 27, last paragraph: remove contraction in first sentence.

It would be helpful to see more concrete recommendations related to future research and how those findings can be used to inform programs, particularly in light of how the paper has been framed.

Can more attention be given to the discussion of limitations in the discussion section? For example, a discussion of transferability and context might be important to include.

Conclusion: I would suggest being careful saying “significant differences” when not referring to a statistical association.

Reviewer #2: General comment: It is very difficult to follow the article without line numbers.

Authors must clearly mention their objectives and research hypotheses in bullet form.

Page 5–6: Sampling: The authors have explained how they have chosen their sample based on the information available from the data collected in the Pahse B. But they must write about the exact sample size, and they must also clarify that figure one has two parts. The upper part of Figure 1 is also considered by the researchers, or they have used it for extracting their own purposive sample size.

Moreover, based on the lower part of Figure 1, The sample size may be equal to 14 villages * 8 households (as written in the text at page 6) = 112. Then why, in Figure 1, does it seem like there are 58 households?

The authors do not clearly describe their sampling. It is not clearly described.

Page 7: Data Collection

What are the characteristics of a research assistant? their education level, age, or any experience that may validate the data collection?

Page 8: "After the survey, participants completed in-depth interviews called the "Food Interviews", which included information on food practises, behaviours, and beliefs, as well as decision-making around food and nutrition. The food interviews were not reported in this paper and will be reported elsewhere." The author must delete these lines from the article because the readers have no concern for what you have collected if this is not part of the current study. Also, similar things have been written in the method part on page 6. "Therefore, we have chosen to focus on this theme in this paper, and additional results of Phase B are reported elsewhere (paper in progress)."

Similarly, all mentions of "the interview" refer to these pile-sort interview questions. Other pile sort results are reported elsewhere due to space limitations."

The authors should focus on their goals and not on the other sides of their questionnaire design. The reader has no concern with that.

at Page 8: "Interviewers took detailed notes after each interview using the audio recordings, which were discussed in detail with the research team. Each interview was recorded and transcribed by the nine interviewers. The interviews were then reviewed by our research coordinator and sent to an independent party for translation into English. After translation, each transcript was reviewed twice: once against the audio recording by the original interviewer and a second time by a different team member." The author wrote that the data was collected through a tablet, and now each interview is recorded. why? Moreover, how did you ensure the validity of the responses collected after the translation by a third party? And if they can review the translation, then why not translate themselves?

I suggest the authors revise their method, especially the pile sort interview and its use (at page 9 in the data analysis section, second paragraph).

On page 10, "just under 20% of couples were polygamous." Is it about the women also? What does it mean that 11% of women are polygamous? It means these women had more than one husband at a time.

"When interviewing the husband, we specified which wife we wanted him to refer to during the interview." How did you specify that, then? And why did you leave others out to include them in the interview? The polygamous couples may have very different behaviours towards their decision-making as compared to the single (husband and wife) couple.

"Who decides how the money you earn will be used? response: I do not know." How is this response possible? I do not think so. Could it be?

"Who usually decides how your partner’s earnings will be used?" It seems that both men and women in a couple were working to earn money. As a result, the authors must include in Table 1 the occupation or earnings (income) of each partner.

"* DHS Questions:" The asterisk form may be placed after the questions in Table 2. There's no need to put it under the table again.

I could not understand the household numbers (73, 61, 39, etc.) and the wife's and husband's ages mentioned in tables 5, 6, and 7. It is not mentioned in the method section or in the results what the purpose of this is.

Based on the descriptive analysis and the responses of the 58 households, is it possible to generalise the results?

I am not satisfied with the discussion; the authors should focus on the direct perceptions of men and women regarding income spending. Moreover, they should explain what the contribution of the study is to the literature, as gender differences in decision-making are a changing phenomenon over time and across places. Simply considering the perceptions of men and women in a couple is enough to conclude who is the decision-maker in a household.

Best of Luck:

6. PLOS authors have the option to publish the peer review history of their article (what does this mean?). If published, this will include your full peer review and any attached files.

Reviewer #1: No

Reviewer #2: No

---

## [Author Response · Author response to Decision Letter 0]

16 Nov 2023

PLOS ONE

Ref: PONE-D-23-01170

Title: Gender differences in perceptions of “joint” decision-making about spending money among couples in rural Tanzania

Review Received: July 18, 2023

Re-submission Due: Nov 1, 2023

Editor Comments

Thank you for submitting your manuscript to PLOS ONE. After careful consideration, we feel that it has merit but does not fully meet PLOS ONE’s publication criteria as it currently stands. Therefore, we invite you to submit a revised version of the manuscript that addresses the points raised during the review process.

Two major concerns shown by the reviewers are: justification of small sample size and weak discussion. Authors must carefully attend them.

Response:

Thank you for this helpful feedback. We have tried to revise and strengthen the manuscript in response to the comments and found many helpful comments from the reviewers for strengthening the discussion.

Editor Comments 

Row/

Comment # Reviewer # Reviewer Comment Response

2 1 Have the authors made all data underlying the findings in their manuscript fully available?

Reviewer #1: No

Reviewer #2: No

 The Tanzania Food and Nutrition Centre (TFNC) hold the ownership rights to these data. Researchers wishing to access it need to obtain written permission from TFNC (info@tfnc.go.tz or https://www.tfnc.go.tz/contactus). Once this is obtained, the lead author on this paper will provide the data.

3 1 Overall: this manuscript incorporates both a quantitative and qualitative exploration of household decision-making as part of a larger project in Tanzania. A comparison of spouses’ responses led to a robust discussion of the complexities of decision-making and the development of a spectrum that can be helpful in understanding household decision-making. Overall, this is an important topic and one that those working in gender are continuing to struggle with. At the same time, there are ways that the manuscript can be strengthened – from the framing in the introduction to specifics in the methods and results, to better contextualization of results and interpretation in the discussion – that would help to strengthen the manuscript overall. Please see more detailed comments below.

 Thank you! We have used both reviewers’ comments to strengthen our manuscript.

4 1

5 1 In the beginning part of the abstract as well as at the end, I wonder if it is more compelling to explain why, for health and wellbeing, it is important to explore dynamics around decision-making. Thank you for this suggestion! We have tried to add this a bit to the abstract while still staying within the short word count.

6 1 The second sentence of the abstract is run-on and it needs to be separated/revised.

 Thank you, we have changed this on page 2.

7 1 Suggest removing “very” from sentences as it is a vague descriptor.

 We have removed “Very” on page 2 in the abstract

8 1 Based on what is written in the abstract, it is unclear how the complexities in decision-making led to the conclusion that couples/households are on a spectrum from no cooperation to full cooperation.

 We have removed that sentence as the complexities in decision-making are addressed in detail later in the paper as there is not enough space in the abstract to fully explain all of the results.

9 1

10 1 Paragraph 1: Can you expand/elaborate, give specifics on what conflict and cooperation about in households can look like?

 We do so later in the introduction in lines 54-56 & 79-96. 

11 1 Paragraph 1, line 4-5: Positive behaviors such as?

 We have clarified that we are discussing health and nutrition behaviors throughout the introduction, specifically in lines 53-59.

12 1 What development objectives does ignoring gender dynamics hinder?

 We have expanded on this in lines 54-56, 79-96, and 100-101

13 1 Is the issue defining gender equality, or implementing and putting into practice and contextualizing what gender equality looks like?

 Both are challenges, as they are tightly connected. We have clarified that both are important on page 3.

14 1 Paragraph 2: women’s empowerment is related to gender equality, but distinct. I think the relationship between these concepts could be clarified. In addition, a lot of different concepts are covered in this paragraph, and how they relate, in particular how/why decision-making is important for improving gender equality, should be expanded.

 We have added that all these concepts include decision-making at the center on page 3.

15 1 I think the organization of the introduction could be improved to strengthen and improve clarity. I think it might be stronger to start the introduction making the important linkage between decision-making and health outcomes, which currently comes in paragraph 3, and why we should understand decision-making and then introduce key concepts. That might help the flow as the introduction currently goes back and forth between gender equality and decision-making. It might also be important to discussion the distinction between women’s decision-making and joint decision-making a bit earlier before page 4. We appreciate this comment and have aimed to reorganize the introduction section. We realize we are tackling many different related concepts, so we have ensured all of the relevant pieces are there.

16 1 Page 5: “Research have found” – in what contexts? In addition, a lot of studies are summarized in this paragraph, and it might help to add a topic sentence or concluding sentence that summarizes/synthesizes the overarching key point in how these studies vary.

 The paragraph explains some of the different studies and the context after that sentence, (e.g., Guatemala, South Asia). We have reworded to make this clearer on page 5.

17 1 It might be helpful to, in the last paragraph of the introduction, contextualize the aims and clarify the location of the study.

 Thanks for this suggestion, we have added a sentence including these on page 6.

18 1

19 1 In the first paragraph, when contextualizing the larger project, it might be helpful to describe briefly the way that Phase A/B results were used as part of the project – were they formative, evaluative, etc.? Phase B focused on household-level attitudes and practices related to what exactly? What nutrition and health behaviors in particular?

 Thank you for this comment. As recommended by the second reviewer, we have simplified the methods and removed all information that is not directly relevant to this study. We have also aimed to clarify that this research focused on decision-making specific to spending.

20 1 It would be helpful to introduce the factors shown to be associated with decision-making in previous research earlier in the introduction if they were considered as part of sampling (e.g., age, education, etc.).

 We have clarified that we wanted to include households with a range of demographics to get diversity in different types of families, not that they were necessarily considered to be a known factor in decision-making. Because decision-making differs so much by context and type of decision, we thought it would not be appropriate to include this (e.g., a paper that found that education was correlated with decision-making for health care). We have also included a citation for our sampling methods used (lines 134-135).

21 1 There is a lot of detail included about the overall study, but I wonder if detail about data collection that is not relevant to the current manuscript should be removed to focus specifically on the methods that led to the data analyzed here.

 Yes, thank you! We have done this for the methods section.

22 1 Data analysis approaches for quantitative data should be included in the data analysis section as well. How was concordance assessed?

 We have provided more information (on line 178-183) on how concordance and discordance was assessed.

23 1 In addition to the analysis of the qualitative transcripts, were particular approaches used to analyze the pile sort activity data? I know space considerations are important to consider, but the results seem to be important to include.

 Since we do not discuss the results of the pile sort in this paper, we have removed it due to space considerations and restricted the methods to what is discussed in this paper.

24 1

25 1 Suggest being specific in the presentation of results rater than saying “about.” For example, not “about 8 years” but presenting the specific number.

 We have removed “about” in the beginning of the results section.

26 1 Any reflections on men’s reflections on DM with one wife vs. another?

 This is a great question. As we discussed in the paper on page 7, we interviewed husbands and wives at the same time (so we specified the wife that was being interviewed while the husband was being interviewed). We had a list of women in our target villages, so we recruited for our study based on the woman. Due to the nature of the interview, we had to specify a particular wife (e.g. “what would happen if you and your wife were to disagree on spending….?” Questions that were more general to the household as a whole included everyone. The issue of polygamy does bring an interesting point that we have expanded on in the discussion.

27 1 p. 10, 2nd paragraph of results: this reads more like interpretation and should be better placed in the discussion.

 We appreciate this comment. However, this paragraph (starting on line 217) is needed to explain why there are differences in reporting between men and women on basic demographic data. We have found that without this section, Table 1 is confusing for readers.

28 1 Given the small sample size, I think presenting statistics with 1 decimal point seems more appropriate. This would make Table 1 consistent with Table 2.

 Thanks for this catch. We now consistently use 1 decimal place throughout the paper.

29 1 The summary of concordance/discordance in Table 3 (and Table 4) could be elaborated in the narrative. Furthermore, it is important to look not only at whether they were concordant or not, but also what that concordant response was. Can that be broken out/disaggregated further? For example, were they concordance that men made the decision, that women mad ethe decision, etc.?

 Thank you for this comment. The breakdown of responses to Table 3 was included in Table 2. Table 4 already specifies the percent responses across all answers, as well as concordance and discordance. We have combined Table 2 and 3 (now Table 2) to make this clear and parallel to Table 4 (now Table 3).

30 1 In describing the example households, it might be helpful to provide more background details about the household (perhaps from the demographic characteristics?). I would also perhaps develop a name for the households rather than using labels like “household 73.” In the tables with quotes from participants, the title of the tables seems to be the nickname/label that could be used. However, it’s important to make this consistent with the sub-heading title where you describe that/those households. Currently, they are slightly differently described.

 We have included age as a demographic characteristic with the quotes, as age difference is an important factor. We are unable to just use the labels because some correspond to more than 1 household (for example, Household 61 & 39 fall into the same category “Conceders and dominators”). However, we have added the label in addition to the household number(s) and relabeled the households using letters A-E for clarity. The paragraphs are organized by how they respond to the question of decision making, not the label we have given them. This is to emphasize the differences in households that would generally respond similarly to a survey question.

31 1 While the tables with illustrative quotes are helpful, is there space to incorporate a few quotes into the main body of the results section?

 Thank you for this suggestion. We have used the quote tables to help readers keep track of the different types of households and to address space considerations in the manuscript.

32 1

33 1 One question that has come to mind in this analysis is that the framing is around household decision-making, but the analyses seem often focused on the husband and wife. Is there space to elaborate on the role of other family members in household decisions?

 Yes, although that is outside the scope of this paper, we agree that other family members involvement is also important. We have expanded on this in the discussion as a limitation of both our study and capturing decision-making in general (paragraph starting at 407). 

34 1 The first two paragraphs in the discussion could perhaps be synthesized with what is in the introduction so that the discussion dives right into the key findings and situating them within the context of existing literature and identifying programmatic implications.

 Thank you for this very helpful idea!

35 1 Page 27, last paragraph: remove contraction in first sentence.

 We have fixed this.

36 1 It would be helpful to see more concrete recommendations related to future research and how those findings can be used to inform programs, particularly in light of how the paper has been framed.

 We have discussed the recommendations for survey measurement and assessment of decision-making in pages 27-28 of the discussion.

37 1 Can more attention be given to the discussion of limitations in the discussion section? For example, a discussion of transferability and context might be important to include.

 Great idea, thanks and done (page 27)!

38 1

39 1 I would suggest being careful saying “significant differences” when not referring to a statistical association. We have removed this word at the beginning of page 29.

40 2 General comment: It is very difficult to follow the article without line numbers. Thank you for catching this. We have added line numbers.

41 2 Authors must clearly mention their objectives and research hypotheses in bullet form.

 We have followed the submission guidelines for both the abstract and introduction (https://journals.plos.org/plosone/s/submission-guidelines#loc-manuscript-organization)

42 2 Page 5–6: Sampling: The authors have explained how they have chosen their sample based on the information available from the data collected in the Pahse B. But they must write about the exact sample size, and they must also clarify that figure one has two parts. The upper part of Figure 1 is also considered by the researchers, or they have used it for extracting their own purposive sample size.

 We have removed the figure and streamlined the text around sampling. We have also removed Phase information and all information not relevant to this study.

43 2 Moreover, based on the lower part of Figure 1, The sample size may be equal to 14 villages * 8 households (as written in the text at page 6) = 112. Then why, in Figure 1, does it seem like there are 58 households?

 We have aimed to clarify in the beginning of page 7 that we chose 4-5 households per village from a list of 8:

44 2 The authors do not clearly describe their sampling. It is not clearly described.

 We have streamlined the text around sampling and clarified how the sampling was done at the beginning of page 7. We have also included the citation for our sampling methods.

45 2 Page 7: Data Collection

What are the characteristics of a research assistant? their education level, age, or any experience that may validate the data collection? Our research assistants had different levels of education, different ages, and different levels of experience. Our team consisted of different people. Some of them had children and some did not. We are concerned that adding t

---

## [Decision Letter · Decision Letter 1]

27 Mar 2024

Gender differences in perceptions of “joint” decision-making about spending money among couples in rural Tanzania

PONE-D-23-01170R1

Dear Dr. Owoputi,

We’re pleased to inform you that your manuscript has been judged scientifically suitable for publication and will be formally accepted for publication once it meets all outstanding technical requirements.

Kind regards,

Muhammad Khalid Bashir, PhD

Academic Editor

PLOS ONE

Additional Editor Comments (optional):

Line 226: Although authors hired research assistants (RA), they can mention their education and age in a range of ways, that may not confuse the readers. and they must discuss the reliability of their data collection.

Reviewers' comments:

Reviewer's Responses to Questions

**Comments to the Author**

1. If the authors have adequately addressed your comments raised in a previous round of review and you feel that this manuscript is now acceptable for publication, you may indicate that here to bypass the “Comments to the Author” section, enter your conflict of interest statement in the “Confidential to Editor” section, and submit your "Accept" recommendation.

Reviewer #2: All comments have been addressed

Reviewer #3: All comments have been addressed

2. Is the manuscript technically sound, and do the data support the conclusions?

Reviewer #2: (No Response)

Reviewer #3: Yes

3. Has the statistical analysis been performed appropriately and rigorously? 

Reviewer #2: (No Response)

Reviewer #3: Yes

4. Have the authors made all data underlying the findings in their manuscript fully available?

Reviewer #2: (No Response)

Reviewer #3: Yes

5. Is the manuscript presented in an intelligible fashion and written in standard English?

Reviewer #2: (No Response)

Reviewer #3: Yes

6. Review Comments to the Author

Reviewer #2: Line 226: Although authors hired research assistants (RA),  they can mention their education and age in a range of ways, that may not confuse the readers. and they must discuss the reliability of their data collection.

Reviewer #3: Authors investigated gender differences in perceptions of“joint”decision-making about spending money among couples in rural Tanzania. The theme of the paper is interesting and important. This manuscript has a clear structure: introduction, methodology, results, and discussion all are well presented. The language is fluent and easy to read. I recommend its publication in PLOS ONE.

7. PLOS authors have the option to publish the peer review history of their article (what does this mean?). If published, this will include your full peer review and any attached files.

Reviewer #2: No

Reviewer #3: No

---

## [Editor Report · Acceptance letter]

8 May 2024

PONE-D-23-01170R1 

PLOS ONE

Dear Dr. Owoputi, 

I'm pleased to inform you that your manuscript has been deemed suitable for publication in PLOS ONE. Congratulations! Your manuscript is now being handed over to our production team.

Kind regards, 

on behalf of

Dr. Muhammad Khalid Bashir 

Academic Editor

PLOS ONE